# Recent Advances and Applications of Rapid Microbial Assessment from a Food Safety Perspective

**DOI:** 10.3390/s22072800

**Published:** 2022-04-06

**Authors:** George Pampoukis, Anastasia E. Lytou, Anthoula A. Argyri, Efstathios Z. Panagou, George-John E. Nychas

**Affiliations:** 1Laboratory of Microbiology and Biotechnology of Foods, Department of Food Science and Human Nutrition, Agricultural University of Athens, Iera Odos 75, 11855 Athens, Greece; georgios.pampoukis@wur.nl (G.P.); alytou@aua.gr (A.E.L.); stathispanagou@aua.gr (E.Z.P.); 2Food Microbiology, Department of Agrotechnology and Food Sciences, Wageningen University & Research, P.O. Box 17, 6700 AA Wageningen, The Netherlands; 3Institute of Technology of Agricultural Products, Hellenic Agricultural Organization DIMITRA, Sofokli Venizelou 1, 14123 Lycovrisi, Greece; anthi.argyri@gmail.com

**Keywords:** sensors, machine learning, rapid methods, food microbiology

## Abstract

Unsafe food is estimated to cause 600 million cases of foodborne disease, annually. Thus, the development of methods that could assist in the prevention of foodborne diseases is of high interest. This review summarizes the recent progress toward rapid microbial assessment through (i) spectroscopic techniques, (ii) spectral imaging techniques, (iii) biosensors and (iv) sensors designed to mimic human senses. These methods often produce complex and high-dimensional data that cannot be analyzed with conventional statistical methods. Multivariate statistics and machine learning approaches seemed to be valuable for these methods so as to “translate” measurements to microbial estimations. However, a great proportion of the models reported in the literature misuse these approaches, which may lead to models with low predictive power under generic conditions. Overall, all the methods showed great potential for rapid microbial assessment. Biosensors are closer to wide-scale implementation followed by spectroscopic techniques and then by spectral imaging techniques and sensors designed to mimic human senses.

## 1. Introduction

Foodborne diseases still remain a difficult issue to resolve with adverse effects both on consumer health and on the economy. According to WHO, foodborne pathogens cause illness to 600 million people and lead to 420,000 deaths, annually [1]. As far as Europe is concerned, The European Food Safety Authority (EFSA) and European Centre for Disease Prevention and Control (ECDC) reported campylobacteriosis as the most frequent foodborne disease, followed by salmonellosis, yersiniosis, Shiga toxin-producing *Escherichia coli*- STEC infections and listeriosis. Listeriosis also had the highest proportion of hospitalized cases of foodborne disease in 2020 [2].

The safety issues associated with food have been mostly based on the presence of pathogenic bacteria, which may lead to health hazards. However, it is noteworthy that according to Regulation 178/2002 of the European Parliament and Commission, a foodstuff is regarded as unsafe not only if it is harmful to consumer health but also if it is not fit for human consumption [3]. Taking this into consideration, spoiled food, which means food with an appearance, taste or flavor leading to its rejection, is also considered unsafe [4].

Considering the above-mentioned, the early detection of foodborne pathogens but also the estimation of microbial load could improve food safety and quality and prevent some of the aforementioned cases of foodborne illness. However, the current, “gold-standard” methods for detecting foodborne pathogens in food samples, although reliable, are also time-consuming and laborious and often restrict food business operators to first releasing their products to market and then having the full microbiological data about a specific batch (especially in fresh products) [5,6].

In 2005, European Commission with Commission Regulation (EC) No 2073/2005 first mentioned rapid methods as “analytical methods other than the reference methods that food business operators should have the possibility to use as long as they provide equivalent results”. Additionally, this regulation created increased needs for microbial testing for specific pathogens and products, for example, *Listeria* spp. in RTE (ready-to-eat) products, thus opening new avenues for rapid detection of microorganisms [7,8].

Presently, there are many different approaches for detecting foodborne pathogens in food samples that differ in terms of ease of use, reliability of results, cost and time effectiveness, etc. Although rapid methods such as spectroscopic techniques have emerged more than a decade ago, there are still some challenges to fully replacing or even substituting conventional methods in detecting pathogens [9,10]. The main reasons behind these challenges have to do firstly with the need for trained personnel and secondly with the high—albeit ever-declining—cost of the equipment/analysis of some methods, for example, nucleic-acid based methods [11]. Lastly, it should be noted that another challenge is the time effectiveness of rapid methods which sometimes is not as proposed (i.e., real-time) but lower due to the need for preliminary steps such as enrichment to reach microbial populations such as >10^3^ CFU/mL [12].

The methods included in this review were: (i) spectroscopic techniques, (ii) spectral imaging techniques, (iii) biosensors and (iv) sensors designed to mimic human senses. Special emphasis was given to data analysis, which is really important for the methods’ interpretability, interoperability, robustness and ease of use. To our knowledge, there is no review in the literature that is up-to-date and includes a broad range of different methods. The purpose of this review was to summarize the recent progress in rapid methods of foodborne pathogens’ detection and quantification. Some applications that have been mentioned below may refer to microorganisms other than foodborne pathogens (e.g., spoilage monitoring) as improper levels of these can render a food inappropriate for human consumption, but also due to the potential of the applied methods to be used in detecting foodborne pathogens in the future.

Although we acknowledge that there are reliable, rapid methods other than the ones mentioned in this review (e.g., nucleic-acid-based techniques), we focused on spectroscopic, spectral imaging and (bio)sensor-based techniques. This was due to their novelty, and their relatively low cost of analysis but also due to their ever-increasing capabilities, which go in parallel with the recent advances in data analysis methods.

## 2. Data Analysis

### 2.1. Multivariate Statistical Analysis

Data analysis is a cornerstone step for all the methods examined in this review due to their ability to collect large, detailed and firstly incomprehensible datasets. The complexity of translating these data arises from the fact that the measurements obtained by different sensors include various sources of variability, thus creating the need for analyzing data with alternative statistical methods. Multivariate statistics is based on the simultaneous analysis of multiple dependent variables against multiple independent variables, therefore providing a more complete view of the obtained data and the sources of variability in one run [13]. When the measurements include long datasets, the typical multivariate analysis includes two steps: (i) the data pre-treatment and (ii) the modeling step. [14]. The most used multivariate techniques for analyzing “biochemical fingerprints” are: principal components analysis (PCA), cluster analysis (CA), linear discriminant analysis (LDA) and partially least squares (PLS) [15].

### 2.2. Machine Learning

Machine learning is a sub-category of artificial intelligence that can develop predictive models based on experimental data (i.e., training data, e.g., sensors’ measurements) in order to make predictions/decisions without being programmed for each prediction/decision [16]. Machine learning is a data-driven approach, thus being really useful when it comes to analyzing sensors’ results that are high-dimensional. The most widespread machine learning algorithms are: support-vector machine (SVM), k-means, algorithms based on decision trees (i.e., random forests, extra trees) and artificial neural networks (ANN) [17]. While multivariate statistics is considered a subfield of statistics, machine learning is considered a subfield of computer science and artificial intelligence. Even in different fields, the two concepts nowadays usually overlap due to their capability of analyzing high-dimensional datasets either focusing more on the underlying interactions between components (multivariate statistics) or on the algorithms and their predictions (machine learning). Consequently, both approaches are of high interest for the methods of this review.

## 3. Spectroscopic Techniques

Spectroscopic techniques employ electromagnetic radiation in order to obtain useful information about the chemical (and thus, the metabolic) profile of the tested sample. The basic categorization of spectroscopic techniques differs depending on the wavelength region of the electromagnetic spectrum (i.e., gamma, x-ray, ultra-violet, visible, infrared, microwave and radiowave), the type of the interaction between the electromagnetic radiation and sample (i.e., absorption, emission, refraction, scattering, resonance, etc.) and the type of the material (i.e., atoms, molecules, nuclei, etc.) [18,19]. The most common spectroscopic techniques used in food microbiology are: fluorescence spectroscopy, terahertz (time-domain) spectroscopy, laser-induced breakdown spectroscopy, Raman spectroscopy and Fourier-transform infrared spectroscopy [20,21,22,23].

### 3.1. Fluorescence Spectroscopy

#### 3.1.1. Background

Fluorescence spectroscopy can be considered a type of emission spectroscopy. In emission spectroscopy, electromagnetic radiation brings atoms or molecules into the excited state and then follows a transition of electrons from excited to lower energy states which causes the emission spectrum (the spectrum of frequencies of the emitted electromagnetic radiation). The discrimination capability resides mainly on molecules (typically aromatic molecules), which in their excited states (prior to the emission phenomenon) emit fluorescence [24,25]. The standard setup used is a spectrofluorometer, which is consisted of: (i) a light source, (ii) a monochromator and/or filter for selecting the emission wavelengths, (iii) photosensitive detectors and (iv) a data analysis platform. The samples with known fluorescent “fingerprints” are placed along with unknown samples in the spectrofluorometer. In short, as light passes through the sample the detected fluorescence signal is converted to digital values which construct a graph. This graph is then used to predict the microbial populations of the unknown samples through data analysis [26].

#### 3.1.2. Applications in Microbiology

Pu et al. (2013) applied fluorescent spectroscopy in order to predict meat spoilage. This study developed fluorescent “fingerprints” (at an excitation wavelength of 340 nm) of meat samples stored at 4 °C and 15 °C. The results were then processed with Multivariate Curve Resolution with Alternating Least-Squares (MCR-ALS) to provide quantitative results that were correlated to fluorescence changes and were attributed mainly to NADH content [27].

Lu et al. (2017) designed an automated device for the on-site detection of foodborne pathogens through fluorescence signals (Table 1). The developed device utilized a light-emitting diode (LED) to induce fluorescence in food samples, which was then measured through a spectrometer. The system also included a rotational stage able to change automatically the samples, along with software for data management. The approach of the study was based on Xu et al. (2017) and included preliminary steps for the separation of pathogens (*E. coli* O157:H7, *L. monocytogenes* and *S.* Τyphimurium) from the samples using immune-magnetic nanobeads followed by conjugation with quantum dots but with different emission wavelengths of 528, 572 and 621 nm [28,29]. Lu et al. (2017) determined a linear relationship between the pathogens’ concentrations and the fluorescence intensity within a broad range of different concentrations [28].

Huang et al. (2019) developed a method that involved fluorescence spectroscopy, although not entirely based on that. This method combined immunomagnetic separation (IMS) and the fluorescent lateral flow immunoassay (FLFIA) able to detect and quantify *E. coli* O157:H7 in milk samples. The method also utilized fluorescent magnetic nanobeads (FMNBs), which are immune-magnetic nanobeads conjugated with quantum dots as a way to combine IMS and FLFIA in one nanostructure, therefore avoiding extra preliminary steps. The FMNBs were used to capture *E. coli* O157:H7 from milk samples and the fluorescent signals were measured. A linear relationship between fluorescence intensity and populations of *E. coli* O157:H7 was assessed with a limit of detection (LOD) of 2.5 × 10^3^ CFU/mL [30].

Courrol and Vallim (2021) analyzed the differences in fluorescence between chicken meat contaminated with *Salmonella enterica* and non-contaminated chicken meat in order to obtain insights that could be utilized in future applications. The methodology of the study was based on the simultaneous acquisition of the fluorescence spectra of boneless chicken breast samples that were spiked with different concentrations of *Salmonella enterica* and the fluorescence spectra of non-spiked samples. The differences in the obtained spectra were attributed mainly to the presence of free NAD(P)H and coproporphyrin fluorescence in spiked samples. The study determined that this information along with other relevant phenomena could be utilized in the rapid detection of *Salmonella* in chicken meat samples [31].

### 3.2. Terahertz Spectroscopy

#### 3.2.1. Background

Terahertz spectroscopy assesses the properties of a matter under an electromagnetic field, based on the interaction THz radiation-matter. The THz radiation is a non-ionizing electromagnetic wave that refers to the region between microwaves and far-IR (THz gap) and is often defined with a band of frequencies ranging from 0.3 to 3 THz or wider from 0.1 to 10 THz (i.e., wavelengths ranging from 3.3 to 333.6 cm^−1^) [20]. Due to the fact that THz radiation is non-ionizing, it can be employed in food or microorganisms without causing unwanted complications or damage, unlike other forms of radiation such as X-rays radiation. The interaction between THz radiation-matter can act as a fingerprint and reveal differences between molecules due to the vibrational and rotational modes of many molecules in the THz region [32]. Terahertz technology bases its detection capabilities on the excitation of low frequency molecular vibrations during THz radiation. It detects mainly weak, intermolecular interactions like hydrogen bonds, Van der Waals forces and hydrophobic interactions. The most important advantages of Terahertz against other regions (e.g., infrared region) are the better penetrating capacity against cell tissues and the lower absorption of water molecules (compared to far infrared) [33].

#### 3.2.2. Applications in Microbiology

Terahertz time-domain spectroscopy (THz-TDS) is a recently developed tool, which uses Terahertz technology and has great potential in food microbiology [20,32,34]. In simple terms, Terahertz time-domain spectroscopy utilizes, in addition to weak interactions, the time factor to reveal time-resolved dynamics that may hide unique structural and dynamic properties (which are overlooked in the case of other types of spectroscopy) [34].

Yang et al. (2016) examined the potential of THz-TDS to detect *S. aureus*, *E. coli*, *P. aeruginosa* and *A. baumanii* (Table 1). The process included samples of pure colonies and heat-treated samples in order to assess if the method could discriminate between dead and alive cells. It also included freeze-dried samples because water seemed to be a significant factor that can interfere with the discrimination capability. The results, after Fourier-transformation of spectral signals, showed that absorption coefficients differed between various intracellular water contents and the ratio between bulk and hydration water. These attributes, according to authors, could be used as indicators of metabolic statements and therefore to be utilized as discrimination factors for alive/dead cells and for bacterial species identification. Yang et al. although claimed the potential of THz-TDS for a label-free, cost-effective and rapid detection of common pathogens, they also indicated some critical gaps that need to be filled in the future [34].

Hindle et al. (2018) employed Terahertz spectroscopy to develop a rapid method of microbiological quality assessment of refrigerated salmon fillets packed under a protective atmosphere (100% N_2_). The method used hydrogen sulfide (H_2_S) of headspace gases as a metabolite-indicator produced by spoilage bacteria. Headspace gases were analyzed by both Terahertz spectroscopy and Selective Ion Flow Tube Mass Spectrometry (SIFT-MS). SIFT-MS was used as a validation method and as a basis for the development of a quantitative prediction model. The results showed a promising capability of THz spectroscopy to detect hydrogen sulfide as a spoilage indicator with a limit of detection (LOD) of 220 ppb (at 400–500 ppb of H_2_S spoilage could be easily detected in a sensory evaluation). Authors highlighted that this method could also be applied to other Volatile Organic Compounds (VOCs) for a more holistic approach. Finally, for a future perspective, it was proposed that this technology could be integrated into packages by constructing sensors that use silicon-based fabrication processes [35].

It should be noted that for more robust applications, THz spectroscopy should be combined with other technologies and materials so as to increase the sensitivity of the approach. In this context, most applications for foodborne pathogens include the integration of samples to metasurfaces and nanomaterials as a way to enhance the interaction THz radiation-sample and microfluidic techniques to reduce the absorption of water in THz region [36]. The aforementioned approach is usually implemented in a lot of studies through biosensors with the use of bioreceptors such as aptamers and antibodies and the measurement of the THz spectra, followed by data analysis [36,37,38].

### 3.3. Laser-Induced Breakdown Spectroscopy (LIBS)

#### 3.3.1. Background

Laser-Induced Breakdown Spectroscopy (LIBS) is an atomic emission technique able to detect spectral fingerprints of various chemical elements by analyzing the UV, Visible and IR emissions in laser-generated sparks [39]. The process includes high-power, laser-generated sparks as the excitation source. When they come in contact with the sample (gas, liquid or solid targets), their energy is partly converted into heat. Therefore, the temperature of the sample increases to form a high-temperature plasma that vaporizes a small amount of material. The produced plasma excites the sample’s constituents and emits radiation before it finally decays to emit an element-specific radiation. A part of this emission (spectral fingerprint) is collected with detectors such as a modified optical fiber bundle and is analyzed to predict the relative abundance of all elements in the sample [39,40,41,42].

#### 3.3.2. Applications in Microbiology

Marcos-Martinez et al. (2011) developed a method for the identification of *Pseudomonas aeroginosa*, *E. coli* and *S.* Τyphimurium isolates, based on Laser-Induced Breakdown Spectroscopy (LIBS) and Neural Network (NN). Initially, a reference spectral database was created for the isolates of the aforementioned microorganisms grown in three different agar media. The spectra were collected in the range of 200–1000 nm. Then, a three-layer Neural Network model was created by using a back-propagation (BP) algorithm for the learning process (the process that optimizes the weight of the output of a previous layer of the Neural Network to the next), tested with common metrics such as sensitivity and specificity and was externally validated. The method demonstrated 100% correct (independent of the culture media) identification for both known and unknown samples; however, the small sample size used for the development of the method was highlighted [43].

Liao et al. (2018) developed a method for qualitative and quantitative detection of four bacterial species/strains (incl. 1 *Staphylococcus aureus*, 1 *S.* Τyphimurium and 2 *E. coli* strains) based on the combination of Three-Dimensional Surface-Enhanced Raman Scattering (3D SERS) and Laser-Induced Breakdown Spectroscopy (LIBS) (Table 1). 3D SERS is a spectroscopic method based on the capture of the inelastic scattering of molecules that are located near metal nanostructures (more details about Raman Scattering in Section 3.4. Raman spectroscopy). The 3D SERS method was used for qualitative detection, whereas LIBS was used for quantification. The methodology included in situ synthesis of settled and free Ag nanoparticles (the nanostructures mentioned above) to form a colloidal bacterial suspension and to create effective nanogaps between them that are necessary to enhance the electromagnetic signal. The spectral data were obtained from the natural evaporation of a droplet of the aforementioned suspension and the identification of bacteria through 3D SERS was attributed to the different spectral properties of cell walls. The spectral data were analyzed with Principal Components Analysis (PCA) followed by Hierarchical Cluster Analysis (HCA) and provided correct classification in all cases. The LIBS method was then applied for the spectral region of 200–800 nm and the most intense atomic emission line was selected for quantification, which was at 279.5 nm and corresponded to the intracellular magnesium ions. For this method, the spectral data were analyzed by fitting the emission lines to Voigt profile (a probability distribution) to reduce noise and then by applying log-log linear regression between fitted peak area and bacterial concentrations to make quantitative predictions. The coefficient of determination R^2^ was determined >0.97 and the limit of quantification (LOQ) was estimated at about 5 × 10^3^ CFU/mL. It is notable that the method was also applied in various water samples with an approximate total duration of 30 min and provided also reliable results (relative standard deviation < 14.9%) within the range of 5.8 × 10^3^ CFU/mL to 5.8 × 10^7^ CFU/mL [44].

E. Yang et al. (2020) developed a method for the identification and quantification of *S.* Typhimurium based on Elemental-Tags Laser-Induced Breakdown Spectroscopy (ETLIBS) (Table 1). The procedure included DNA aptamers as elemental tags, silicon nanowires (SiNWs) modified by metal nanoparticles to form a substrate and Laser-Induced Breakdown Spectroscopy for the analysis. For the modification of the silicon nanowires (SiNWs), a mixture of gold and silver (Au@Ag) nanoparticles (Au@AgNPs) was added and the SiNWs-Au@Ag substrate was created. The elemental tags were created by assembling copper nanoparticles (CuNPs) with ssDNA oligonucleotides (aptamer for capture and poly-T sequences as template for CuNPs) and were then incubated with *S.* Typhimurium suspensions for 30 min to achieve labeling. The labeled samples were then added to the SiNWs-Au@Ag substrate to create a complex of SiNWs-Au@Ag/*S.ty*/CuNPs that was used for the LIBS analysis which lasted about 5 min. After the complex was created, the point was to follow Cu peaks in order to obtain information about the sample. The use of Cu as the indicator for *S.* Typhimurium prevalence and quantification was evaluated and established. Subsequently, the spectral data (emission lines) were first fitted to Voigt profile to reduce noise and then log-log linear regression between fitted peak area and bacterial suspensions was used to develop a quantitative model. The coefficient of determination (R^2^) was 0.978. The limit of detection (LOD) was estimated at 61 CFU/mL at a range of detection from 10^2^ to 10^6^ CFU/mL. The capability of this method was tested with suspensions of spiked samples with background microbiota present and by comparison with qPCR. The method achieved *S.* Typhimurium recoveries >87% and relative standard deviations (RSD) <11.56% at spiked samples, in all cases. In conclusion, it should be mentioned that this method could possibly integrate other pathogens with different aptamers for simultaneous identification and quantification; however, it was not yet tested with food samples that were naturally containing *S.* Typhimurium but with food samples’ dilutions (made with ultra-pure water for 12 h) that were artificially spiked [45].

### 3.4. Raman Spectroscopy

#### 3.4.1. Background

Raman spectroscopy is a vibrational spectroscopy method that may take part as a useful tool in food microbiology since it is non-destructive, requires minimal preparation of samples, can be integrated into a portable device and can provide information about different molecules, simultaneously [46].

Raman spectroscopy source its name from the Raman scattering or Raman effect that was experimentally observed by Raman and Krishnan in 1928, nevertheless the starting point of the method was after 1960 with the evolution of technology [47]. In short terms, in Raman spectroscopy, a sample is targeted and exposed to a monochromatic light, which leads more to the transmission of light through the sample, less to the scattering of light with the same wavelength and even less to inelastic scattering of light with a different wavelength. The last-mentioned effect is known as Raman scattering and although the least in terms of scattering photons (about 1 inelastic-scattered photon out of 10^6^–10^8^ incident photons), it is the most important. This is because these photons, when falling in a sample, cause some molecules to move at a different rotational/vibrational state and also the scattered photons to move at a different state (Raman shift). The energy difference between the two states of photons is equal to the energy difference between the two vibrational states of the molecules and can therefore provide useful information about the molecules in a sample [47,48,49].

Biological samples like foods or microorganisms’ suspensions contain a lot of different molecules, so the Raman spectrum arises mainly from the spectral superposition of the molecules within the monochromatic light and serves as a spectral fingerprint i.e., a reference for the microbiological interpretation of spectral results after data analysis [48,50].

#### 3.4.2. Applications in Microbiology

Argyri et al. (2013) demonstrated for the first time that Raman spectroscopy can become a useful tool for the rapid assessment of meat spoilage. The method was based on implementing the direct analysis of meat samples with Raman spectroscopy along with simultaneous, conventional microbiological analysis for the development of a quantitative model after data analysis for the correlation of the results. At the same time, the pH changes in meat samples, as well as their organoleptic acceptance during preservation, were also monitored, and a qualitative (semi-quantitative) model with three classifications (fresh, semi-fresh, spoiled) was developed. Data analysis was conducted through advanced multivariate statistical methods and machine learning methods. The developed models were validated with half-out cross-validation. In conclusion, the quantitative results were promising regarding the time needed for the direct analysis with Raman Spectroscopy. Specifically, radial basis function of support vector machines regression (RBF) (SVR_R_) and sigmoid functions of support vector machines regression (SVR_P_) gave acceptable predictions ($\%\;\mathrm{PE}=\left(\frac{\mathrm{PEin}}{\mathrm{PEtotal}}\right)\times \;100>70\%$) for all of the counts. However, genetic algorithms-artificial neural networks (GA-ANN) gave better results considering the fact that no fresh sample was misclassified as spoiled and vice versa [46].

Lu et al. (2020) developed a label-free method that combined artificial intelligence and Laser Tweezers Raman Spectroscopy (LTRS) for the identification of 14 microbial species [50] (Table 1). Briefly, LTRS is a Raman spectroscopy variant that is used to analyze single cells and biological particles suspended in an aqueous environment [51]. Lu et al. (2020) used a Raman Tweezers System that combined an optical trap, a confocal microscope, a Raman spectroscope and other optical elements in order to analyze single cells (as discriminated and determined by the optical trap and the confocal microscope) at different growth states of 14 microbial species. The full Raman spectra (defined as ramanome) of each microbial cell were collected and analyzed with a convolutional neural network (ConVet). For better investigation of the ConVet results, they developed the Occlusion-Based Raman Spectra Feature Extraction (ORSFE) tool which provided the ability to visualize the Raman features that contributed most to the classification. The results determined correct classifications of the 14 microbial species with an average accuracy of classification at 95.64 ± 5.46%, although it should be mentioned that the current, high cost of the confocal microscopy instrumentation may limit the applications, especially if this method is modified to find food microbiology-related applications [50].

### 3.5. Fourier-Transform Infrared Spectroscopy (FTIR)

#### 3.5.1. Background

Fourier-Transform Infrared Spectroscopy (FTIR) is an analytical, vibrational spectroscopy technique that is able to obtain information about the rotational and vibrational transitions of molecules based on their infrared spectrum of absorption and emission. In the FTIR, a full spectrum beam of IR radiation is passed through the Michelson interferometer. Michelson interferometer is an array of mirrors, one of which is moved by a motor. As this mirror moves, some wavelengths of light in the beam are periodically emitted while others are blocked, providing different recombination of the initial beam at each time [52,53,54]. Every recombination contains multiple, predetermined wavelengths of light and the emitted spectrum is targeted to the sample where absorption and emission are calculated and indicate some vibrations within the molecules (changes at bond lengths (stretching) or changes at the bond angles (bending)). The process is repeated multiple times and the collected data consist of the interferogram. These data show the absorption for each movement of the mirror and not the absorption for each wavelength, therefore needing a conversion, which is made with the Fourier-transform algorithm [52,55,56,57].

The above features were utilized in several applications in food microbiology where FTIR was used as a biochemical fingerprint technique which when combined with data analysis provided useful insights related to food safety and quality [58]. The main point of this approach is that when bacteria are growing in food or other substrates are producing certain metabolites (biochemical fingerprint) that can be detected with FTIR and then be correlated with microbial populations to provide qualitative or quantitative assumptions [59].

#### 3.5.2. Applications in Microbiology

Fengou et al. (2019) evaluated FTIR for its capability of estimating fish microbiological quality. The methodology was based on the approach of Argyri et al. (2010) (biochemical fingerprinting) that was mentioned above and included FTIR measurements of fish samples during preservation at different temperatures and at different time points [60,61]. FTIR results were then correlated with total viable counts (TVC) (as determined by conventional microbiology methods) to provide quantitative predictions through PLS-R models. The FTIR spectra that were mainly utilized were in the range of 3100 to 2700 and 1800 to 900 cm^−1^ and the developed model was validated with leave-one-out cross-validation. The results showed that FTIR could be a useful tool for predictions of TVC in fish samples (both whole and fillets) and it is indicatively mentioned as a metric that the root mean square error (RMSE) of the developed model was estimated at 0.717 log CFU/g [60].

Under the same point of view, Spyrelli et al. (2021) evaluated FTIR as a method for assessing spoilage on the surface of chicken breast fillets. The method involved the analysis of samples with FTIR in various temperatures and time points until spoilage. Simultaneously, the samples were analyzed with conventional microbiology methods for TVC and *Pseudomonas* spp. enumeration. After FTIR analysis, spectral data were first smoothed with the Savitzky–Golay filter for noise reduction of spectra and then the range of 900–2000 cm^−1^ was utilized for correlation with TVC and *Pseudomonas* spp. populations through partial least squares regression (PLS-R) models. A second approach for data analysis was also implemented with the testing of nine machine learning algorithms in the range of 800 to 4000 cm^−1^. In both cases, the developed models were validated and metrics for predictions’ performance such as RMSE were estimated. The results claimed reliable quantitative predictions for TVC and *Pseudomonas* spp. at chicken breast fillets through FTIR with the PLS-R model but also with some machine learning algorithms. It is indicatively stated that the least-angle regression (lars) model achieved quantitative predictions of TVC in samples of independent batches with an RMSE of 0.851 log CFU/cm^2^ [61].

For the detection of foodborne pathogens, Y. D. Wang et al. (2017) combined synchrotron radiation-based FTIR (SR-FTIR) microspectroscopy to develop a method able to identify and discriminate 10 foodborne bacteria (including *Salmonella* spp. and *Vibrio* spp.) (Table 1). SR-FTIR microspectroscopy is an FTIR variant that can analyze samples at the micron level, therefore, providing higher signal-to-noise, precise results in a manner of detecting differences between bacteria at the single-cell level. Y. D. Wang et al. utilized the advantages of SR-FTIR in the analysis of bacterial suspensions. They created a database of spectra (both full spectra and subdivided into regions) for the bacterial suspensions of the ten pathogens and then used Principal component analysis (PCA) to develop the identification and discrimination pattern. The results showed that full spectra could provide better identification and discrimination of foodborne pathogens and in fact, it could even discriminate species between the same genus. However, it should be mentioned that this method was not tested against the background microbiota of food samples or mixtures and also involved preliminary steps that therefore increased the total time of the analysis [62].

**Table 1 sensors-22-02800-t001:** Most representative applications of spectroscopic and spectral imaging techniques in foodborne pathogens’ detection and quantification.

Technique	Microorganisms	Purpose	Data Analysis	References
Fluorescence spectroscopy	*E. coli* O157:H7, *S.* Typhimurium, *L. monocytogenes*	On-site detection in lettuce samples	Savitzky–Golay filter, WA Multiscale Peak Detection, Linear regression	[28]
THz-TDS	*S. aureus*, *E. coli*, *P. aeruginosa*, *A. baumanii*.	Detection and alive/dead cells discrimination in culture media	Fourier transformation, standard algorithm	[34]
LIBS	*P. aeroginosa*, *E. coli*, *S.* Typhimurium	Detection in culture media	Neural network	[43]
3D SERS and LIBS	*S. aureus*, *S.* Typhimurium, *E. coli*	Direct quantification in water	PCA, HCA, Voigt profile fitting	[44]
ETLIBS	*S.* Typhimurium	Quantification in bacterial suspensions and detection in spiked food samples	Voigt profile fitting, Log-log linear regression	[45]
LTRS	14 microbial species	Discrimination in single cells	Convolutional neural network (ConVet), Occlusion-Based Raman Spectra Feature Extraction ORSFE) tool	[50]
SR-FTIR microspectroscopy	10 foodborne bacteria	Discrimination in bacterial suspensions	PCA	[62]
HSI	*E. coli* O157:H7 and *Staph. aureus*	Quantification in pork samples	Voigt profile fitting, 2nd derivatives, SNV VCPA, IRIV, GA	[63]

## 4. Spectral Imaging Techniques

The term “spectral imaging techniques” can be used to describe all the methods that combine spectroscopy and image analysis. Spectroscopy was thoroughly discussed above, while imaging, in short, is about obtaining spatial and temporal data information from objects through various methods (X-rays, MRI, optical methods, etc.). When the two methods are combined, they can provide dual information acquired from spectra and pixels and specifically the spectrum at each pixel [64]. Spectral images can be two-dimensional (2-D) with a spectral and a spatial dimension, or three-dimensional (3-D) with two spatial dimensions and one spectral dimension [65,66]. Spectral imaging uses wavelengths ranging from the visible spectrum—like images from a conventional camera capture—to the wider electromagnetic spectrum of UV and IR. The difference between multispectral imaging and hyperspectral imaging is that the former uses a low number (usually <20) of specific bands (multiband imaging) in the electromagnetic spectrum, while the latter uses a more continuous approach with often hundreds of different spectral bands [20,65].

### 4.1. Multispectral Imaging (MSI)

#### 4.1.1. Background

Multispectral imaging (MSI) is a type of spectral imaging that is usually performed through wavelength dispersive devices or narrow-band filters to separate lights of different spectral bands while an array detector is used to capture them [66]. So, the main components of a multispectral imaging system are: a light source, a dispersive device and a detector. Briefly, the light is emitted from the light source, it is separated into spectral bands and is then targeted to the sample. One part of the incident light is transmitted through the sample and another one is reflected. The detection capability of the method relies on these two attributes (light transmission and light reflection) and computer vision technologies (e.g., image registration) [67].

#### 4.1.2. Applications in Microbiology

Manthou et al. (2020) utilized MSI to develop a rapid method for the estimation of TVC and sensory attributes in ready-to-eat pineapple samples. The methodology involved MSI analysis of samples that were stored at different temperature conditions until spoilage, while at the same time, conventional microbiological methods for enumeration of the TVC and sensory analysis were applied. MSI data were associated with TVC in three different ways: linear regression, Unscrambler software (i.e., a commercial software product for multivariate statistical analysis, CAMO Software AS, Oslo, Norway) and SorfML (i.e., an automated ranking platform that includes machine learning regression models, www.SorfML.com) to develop quantitative models. Unscrambler and SorfML platforms were also used for qualitative models based on sensory analysis. Although the coefficient of determination (R^2^) was low in quantitative models, RMSE values were under 1.0 log CFU/g indicating a satisfactory model performance. For the sensory features (qualitative models), the results showed potential although further investigation with more balanced data was needed for more reliable conclusions [68].

Following the same concept, Spyrelli et al. (2020) developed a method for rapid quantification of *Pseudomonas* spp. and TVC in different chicken products, through MSI to estimate the “time from slaughter”. The analytical procedure was similar to the described above and for the development of the quantitative model, Partial Least Squares Regression (PLS-R) was used. The results showed that the developed models could effectively predict *Pseudomonas* spp. and TVC in any chicken product (and therefore the “time from slaughter”) and is indicatively mentioned that the developed model for the chicken thigh achieved an RMSE value of 0.160 and a correlation coefficient (r) of 0.859 [69].

Regarding the detection of foodborne pathogens with Multispectral imaging, there are no relevant research works available so far.

### 4.2. Hyperspectral Imaging (HSI)

#### 4.2.1. Background

As mentioned above, Hyperspectral imaging (HSI) shares the same principles with MSI and the difference lies in the number of bands involved. HSI provides higher spectral resolution and lower spatial resolution as a result of the hundred(s) of bands (more continuous approach) measured [70]. The higher number of bands can provide more in-depth details and accurate fingerprints of samples; however, the extra bands may reduce intensity and signal-to-noise ratio. The above characteristics of HSI, as will be discussed in the following section, may provide the capability for foodborne pathogens’ detection, although the increased complexity of data processing may limit the applications in the food industry [71,72].

#### 4.2.2. Applications in Microbiology

Michael et al. (2019) developed a method for rapid differentiation of previously isolated *Cronobacter sakazakii*, *Salmonella* spp., *E. coli*, *L. monocytogenes* and *Staph. aureus* through HSI. The method involved the isolation of different strains of the aforementioned bacteria and the immobilization of them in glass slides which were then analyzed with HSI for the development of a database. The wavelength range of 425.57 to 753.84 nm was selected and then PCA and kNN (k-nearest neighbor) classifier modeling were applied. The developed model was cross-validated, and the results were satisfactory for some strains, while the model’s performance was poor in some others. The authors proposed the use of the method for testing the presumptive presence of foodborne pathogens, but it was mentioned that it cannot yet replace conventional microbiology methods [73].

Bonah et al. (2020) compared variables’ selection algorithms for quantitative monitoring of *E. coli* O157:H7 and *Staph. aureus* in pork samples through visible near-infrared (Vis-NIR) hyperspectral imaging (Table 1). The methodology involved the inoculation of pork samples with the pathogens at certain levels before the acquisition of the spectra with Vis-NIR HIS. The spectral data were preliminarily processed with “noise-reducing algorithms”, including Savitzky–Golay filter, Second derivatives, and Standard Normal Variate (SNV). Afterward, for the determination of representative variables, they compared six different wavelengths’ selection algorithms and their combinations for predictions’ optimization. The algorithms’ predictions were evaluated through various metrics including root mean square errors of: (i) calibration, (ii) cross validation and (iii) prediction on the prediction set. The results indicated that the combination of Variable Combination Population Analysis (VCPA), informative variables (IRIV) and Genetic Algorithm (GA), may be a suitable set of algorithms for quantitative monitoring of foodborne pathogens in food samples through HSI [63].

## 5. Biosensors

### 5.1. Background

Some of the above methods are often combined with (or categorized as) biosensors. In this study, biosensors are considered as methods, which convert easily measurable properties such as optical, electrochemical, magnetic, piezoelectric, etc. to microbiological predictions. On this basis, the typical components of a biosensor are: a detector (which interacts with possible antibodies, nucleic acids, metabolites, proteins, etc. and provides a signal), a transducer (converts this signal to an electronic output, that is, the easily measurable properties that were mentioned above) and a display layout (provides the microbiological prediction) [74] (Figure 1). There are several types of biosensors and more than one categorization method. Briefly, the types with the most applications are electrochemical, optical and mass-sensitive biosensors.

Electrochemical biosensors (amperometric, potentiometric, conductometric, impedimetric) measure changes in electrons or ions, as a result of different reactions between the bio-recognition element (core component) and the captured molecules of the analyte [75,76].

Optical biosensors are devices that use optical transduction and therefore their measurements are based on changes in light emission/absorption and usually changes of the refractive index as a result of the core component-analyte binding [77]. Apart from the refraction, other “light-based” or spectral properties can also act as indicators, thus creating sub-categories such as reflection, resonance, Raman scattering, THz, fluorescence, chemiluminescence biosensors, etc. [76].

Mass-sensitive biosensors, also known as piezoelectric biosensors, “sense” minute changes in mass by featuring materials of fixed mass that accumulate piezoelectricity and vibrate in a specific frequency at a predetermined alternating current. When the mass changes as a result of the core component-analyte binding, so do the vibration of the material and consequently the output signal [76]. The most common biosensors in this category are surface acoustic wave biosensors, resonant-mode piezoelectric sensors and quartz crystal microbalance sensors [74,78].

### 5.2. Applications in Microbiology

Yamada et al. (2016) designed a single-walled carbon nanotube- (SWCNT) based multi-junction sensor and tested it against *Staph. aureus* and *E. coli*. The assumption was based on the linear regression between microorganisms and sensor results, providing a detection range of 10^2^–10^5^ CFU/mL [79].

Another new biosensor was made by Huang et al. (2018) and combined: (i) double-layer capillary-based high gradient immunomagnetic separation, (ii) invertase-nanocluster-based signal amplification and (iii) glucose meter-based signal detection to create a new biosensor able to detect *E. coli* O157:H7, an assumption that was based on the linearity between the signal and a pathogen, with a LOD of 79 CFU/mL [80].

Wang et al. (2019) combined immunomagnetic separation, fluorescence labeling and smartphone video processing to create a microfluidic biosensor for *S.* Typhimurium detection with a LOD (under given conditions) of 58 CFU/mL [81].

Díaz-Amaya et al. (2019) developed an aptamer-based surface-enhanced Raman spectroscopy (SERS) biosensor for the detection of *E. coli* O157:H7. As mentioned above, SERS is a surface-sensitive, Raman spectroscopy variant which was developed in order to enhance the Raman scattering effect to detect single molecules by using metallic nanoparticles. This study combined SERS with gold nanoparticles (GNPs/Au-NPs) conjugated DNA aptamers specialized for E. coli O157:H7 detection. The procedure started with the development of AuNPs-Raman reporter complexes and their mixture: (i) with serial dilutions of the pathogen (pure cultures) and (ii) with dilutes of ground beef samples spiked with predetermined concentrations of the pathogen. Afterward, spectral data of the samples were collected with SERS and analyzed by a One-way ANOVA test, accompanied by means of comparison using Dunnett’s method. The results determined the ability of detection and quantification of *E. coli* O157:H7 with a LOD of 10^2^ CFU/mL for spiked ground beef homogenates [82].

Tsougeni et al. (2020) designed an integrated lab-on-chip platform able to detect *S.* Typhimurium, *Bacillus cereus*, *Listeria* spp. and *E. coli*, separately, in milk samples. The platform was based on a surface acoustic wave (SAW) biosensor combined with antibodies (capture) and isothermal DNA amplification (detection and identification). The protocol involved spiking milk samples at minimal levels (~1–5 CFU/mL), followed by a two-step pretreatment (i.e., preculture of 3–4 h and centrifugation). Then, the steps followed were all integrated into one platform, namely, antibody binding, cell lysis, DNA isothermal amplification and SAW detection. It was demonstrated that the total time of analysis was less than 4.5 h, although the method was not yet validated with solid food samples [83].

## 6. Sensors Designed to Mimic Human Senses

### 6.1. Background

Recent advances in technology have made it possible to simulate the human senses such as smell and taste, for the purpose of sensory evaluation and determination of food quality and safety. The developed instrumental techniques are known as the electronic nose (e-nose) and electronic tongue (e-tongue). These techniques are either based on sensors that are equipped with chemosensitive materials to interact with specific substances or on bioelectronic devices (bioelectronic nose and tongue) with molecular recognition elements that can mimic the olfactory and taste systems [84]. In addition, e-nose and e-tongue (and their bioelectronics counterparts) can assess the VOCs (odorant and malodorous) in a relatively short time and following a simpler methodology compared to more complex systems such as GC-MS [85,86]. Putting things into context, e-nose and e-tongue are not able to replicate the human smell and taste but can gain ground on them by detecting also non-odorant compounds, therefore providing a useful approach with broad applications [85]. An e-nose system is usually constituted by gas sensor arrays (e.g., metal oxide semiconductor (MOS) gas sensors), while chemical sensor arrays (e.g., potentiometric) constitute an e-tongue [87]. E-nose measures the volatiles in the headspace of a sample, whereas the e-tongue detects the non-volatile compounds of a liquid sample [88] (Figure 2).

### 6.2. Applications in Food Microbiology

Astantri et al. (2020) utilized e-nose properties to monitor the growth and discriminate *L. monocytogenes* and *Bacillus cereus* in Tryptone soy broth (TSB). An array of MOS gas sensors was used to monitor changes in VOCs after inoculation of TSB samples with 10^3^–10^4^ CFU/mL of each bacterium. The e-nose measurements were performed every few hours and up to 48 h and were then compared with the results of conventional methods of microbiology. E-nose results showed a distinct pattern between *L. monocytogenes*, *B. cereus* and non-inoculated samples. Among the algorithms used for discrimination, SVM algorithms seemed to provide the best accuracy of prediction. The qualitative results of this study were promising and a causal relationship between specific VOCs and the pathogens was also determined; however, the method was not tested in food samples or enrichment broths [89].

Bonah et al. (2021) developed a method for the detection of *S.* Typhimurium in minced pork samples through e-nose. An array of MOS gas sensors was used to acquire measurements of non-inoculated and inoculated samples of different populations (10^2^, 10^4^ and 10^7^ CFU/g) while conventional microbiological analysis was also performed. Principal components analysis (PCA) was used for qualitative discrimination of inoculated samples, whereas SVM algorithms were used to construct a model for quantitative estimations. SVM regression models were also developed with and without optimized hyperparameters [90]. Hyperparameters are often used in machine learning to configure the initial training process [91]. The results showed that SVM with optimized hyperparameters exhibited good performance and could be used for quantitative estimations of *S.* Typhimurium in pork samples, whereas PCA could be used for qualitative discrimination [90].

Al Ramahi et al. (2019) utilized an electronic tongue to discriminate *E. coli*, *Staph. aureus* and *P. aeruginosa* in nutrient broth. An array of seven potentiometric MOS sensors was used for the analysis of different samples of nutrient broth inoculated with the three pathogens. The results were processed with PCA and it was claimed that the method could discriminate efficiently the three isolates after 15 h of incubation. The main limitation of the research was that the method was tested only against isolates and not in food samples, therefore, inducing uncertainty regarding the applicability of the method [92].

Ghrissi et al. (2021) used e-tongue to discriminate and quantify *Enterococcus faecalis*, *Staph. aureus*, *E. coli* and *P. aeruginosa* in aqueous dilutions. Two arrays of a total of 40 lipid polymeric cross-sensitive sensors were used to capture the potentiometric fingerprints of each sample (isolates). These sensors were considered to interact chemically with the bacterial cell walls, possibly in a different way in each case. Firstly, linear discriminant analysis coupled with a meta-heuristic simulated annealing algorithm for variable selection (LDA-SA) was used to develop a model for the discrimination of the microorganisms. Secondly, multiple linear regression coupled with a simulated annealing algorithm (MLR-SA) was used for the selection of the most suitable sensors’ measurements to develop a model for quantification. Both models were validated with leave-one-out cross-validation (LOOCV). Although the statistical metrics were really promising for the developed models, it is still unsure if the method could be used for applications in food samples or enrichment broths [93].

Carrillo-Gómez et al. (2021) tested both e-tongue and e-nose to determine their capabilities of discriminating *Escherichia coli* from *Klebsiella pneumoniae* and *Salmonella enterica* in pasteurized milk samples. Among the different data analysis methods used in the study, it was determined that for e-nose, the SVM algorithms with Gaussian mean provided the best results in terms of classification (94.7% success rate). For e-tongue, k-nearest neighbors (k-NN) algorithm with fine adjustment achieved the best results (98.7% success rate). The limit of detection (LOD) for *E. coli* was claimed to be 0.01 CFU/mL of milk [94].

## 7. Limitations and Challenges

The limitations and challenges of the rapid methods included in this review have to do either with the technical capabilities of the methods of detecting molecules in really low concentrations or with the following data analysis to “translate” these measurements into valuable insights. The first aspect is constantly evolving with the evolution of technology, whereas for the data analysis, a lot of limitations arise with the inappropriate use of the available statistical and machine learning tools. As mentioned above, the combination of multivariate statistical analysis or machine learning with (bio)sensors could be appropriate to unfold the complex features of the analysis procedure and to develop more accurate models. However, large amounts of data are required to develop models with high predictive power. This is not always the case with the models reported in the literature due to the difficulties of obtaining a high amount of biological data, leading in a lot of cases to confusion between data fitting and modeling or to models that are not adequately validated [95]. Machine learning methods, although promising, come with disadvantages when used improperly. The most important wrong practice when implementing machine learning algorithms to make predictions is “the curse of dimensionality” [96]. This is the result of trying to include a lot of parameters so as to get better predictions, meanwhile leading to the opposite outcome. Data leakage is also another issue and can be described as the case when the training data are not equivalent to test data, leading to overgeneralization. Additionally, limited datasets can lead to data bias and models that represent only a small fragment of total variability, thus providing low predictive power under generic conditions [17]. Finally, with existing means, it still remains difficult, to determine if the source of a variation observed between similar treatment conditions is due to a specific factor that is different or due to the unspecified practices of each analyst/laboratory. Thus, when it comes to the underlying principles of these methods, it is difficult to assert a clear causality (and its extent) from correlation.

## 8. Conclusions (Future Perspectives)

As was determined in previous reviews [23,97,98,99] the need for rapid methods able to detect and quantify foodborne pathogens was increased and so were the proposed alternative methods. This review differentiated from the past in terms of including methods that were previously omitted or overlooked because their potential and applicability were not yet established (e.g., spectral imaging techniques, sensors designed to mimic human senses, etc.). In addition, the methods that were previously examined were again described along with recent applications in order to provide up-to-date insights.

Spectroscopic and spectral imaging techniques, share the same strengths as they are both non-destructive and have low or zero cost of analysis. Spectroscopic techniques seemed to be more precise and closer to application than spectral imaging. However, the spectral imaging measurements are faster, and thus more relevant to online microbial assessment in the food industry. For the spectroscopic techniques it would be interesting to carry out more applications with a combination of the two techniques simultaneously (e.g., SERS with LIBS) or the combination of spectroscopy and biosensors (e.g., THz biosensors) since these combinations seemed to provide more precise results on foodborne pathogens’ detection and quantitative estimation. For spectral imaging, the improvement of sensitivity and specificity would be an important factor for future applications. For both spectroscopic and spectral imaging techniques a future challenge would be the utilization of data analysis (e.g., with multivariate and machine learning approaches) [68].

As a future perspective regarding biosensors, the development of materials and biorecognition systems (e.g., aptamers) should be mentioned. Moreover, the validation of biosensors with in-field applications could attract food industries’ interest due to their lab-on-a-chip approach.

Sensors designed to mimic human senses appeared to be still in their first steps referring to foodborne pathogens detection and quantification. Therefore, more applications would provide more accurate insights in the future into the capabilities of this approach.

In summary, not every one of the aforementioned methods seemed to be at the same distance from “real” applications in common laboratories or food industries. Some of them appeared to be closer to wide-scale implementation like biosensors, others appeared to be still in their infancy such as spectral imaging techniques and sensors designed to mimic human senses, while spectroscopic techniques appeared to be in an intermediate state. A significant drawback of most of these methods had to do with the unavoidable step of pre-enrichment or other preliminary steps and with the data analysis that was not so accurately implemented.

## Figures and Tables

**Figure 1 sensors-22-02800-f001:**
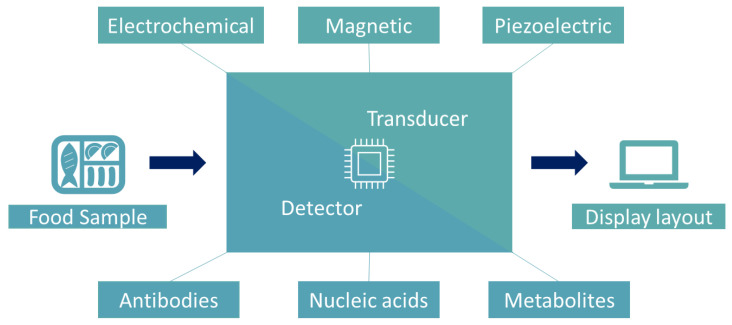
Schematic overview of biosensors’ main components i.e., a detector, a transducer and a display layout.

**Figure 2 sensors-22-02800-f002:**
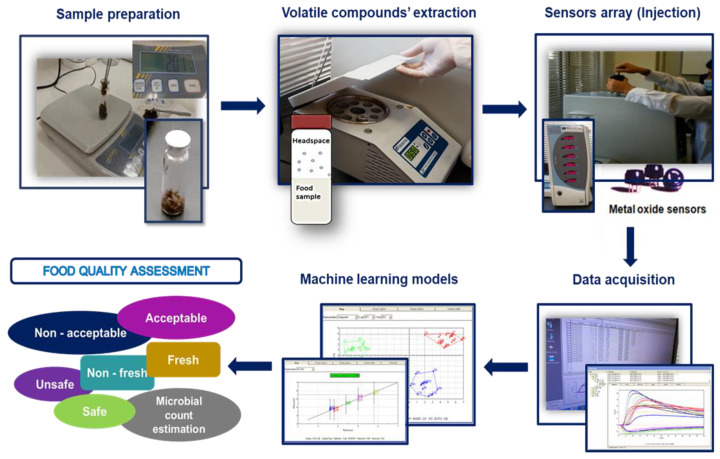
Schematic overview of an e-nose system for rapid food quality assessment.

## Data Availability

Not applicable.

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
