# Peer review of "Recent Advances and Applications of Rapid Microbial Assessment from a Food Safety Perspective"

_sensors, 2022, doi:10.3390/s22072800_

Round 1

Reviewer 1 Report

This paper presents a review on recent advances  for the foodborn disease and food safety assessments, which includes detection techniques and data analysis approaches. The paper is well organized and written and includes a comparison of the most important techniques in terms of the type of microorganisms that can be detected, the goal and required data analysis, such as FFT, neural networks, Raman spectroscopy, etc.

In general, the paper could be considered as is for publication, but it could also be improved by enhancing the quality of Figure 1, in terms of aesthetics and way to convey the idea; and including more figures that can describe the discussion on the techniques. 

Author Response

Response

The authors would like to thank the reviewer for his/her valuable comments.

Figure 1 was revised according to the reviewer’s comments. Please see L568-570

Figure 2 was included in the article as an example to describe all the steps followed for the food quality assessment using a sensor – from sample preparation to data acquisition, model development and finally to the ‘sensors’ response. In this case e-nose was set as an example, any other sensor can be used in a similar manner.

Reviewer 2 Report

Dear Authors,

The reviewed article Ms ID: sensors-1654664 (Recent advances and applications of rapid microbial assessment from a food safety perspective) is  informative about the recent progress towards rapid microbial assessment through  spectroscopic techniques;  spectral imaging techniques;  biosensors; sensors designed to mimic human senses and  focuses on a topic in a way that has not been done before. Moreover the presented descriptive article of investigated  field  has value in terms of summarising the state of the literature.

The paper presents a review of literature  that is relevant in the light of arguments used to support it. Research is very gripping as well as it also  has its scientific value. I think the motivations for this study are very clear. The introduction provides a good, generalized background of the topic that quickly gives the reader an  appreciation of the scientific relevance and timeliness of the research theme. Manuscript is well-written. The presented review article meets an important originality criterion.

However I have some suggestion for Authors to improve further their study. These follow the text sequence:

  • Here are a current reference to consider, Authors can use them in the article:
  1. Tsougeni, G. Kaprou, C.M. Loukas, G. Papadakis, A. Hamiot, M. Eck, D. Rabus, G. Kokkoris, S. Chatzandroulis, V. Papadopoulos, B. Dupuy, G. Jobst, E. Gizeli, A. Tserepi, E. Gogolides, Lab-on-Chip platform and protocol for rapid foodborne pathogen detection comprising on-chip cell capture, lysis, DNA amplification and surface-acoustic-wave detection, Sensors and Actuators B: Chemical, Volume 320, 2020, 128345, ISSN 0925-4005, https://doi.org/10.1016/j.snb.2020.128345

From my standpoint, the presented  manuscript is appropriate for publication Journal Sensors, after minor revision, given the above aspects.

Author Response

Response

The authors would like to thank the reviewer for his/her valuable comments.

The suggested reference was incorporated in the article according to the reviewer’s comment. Please see L 598-607
